# Resveratrol Favors Adhesion and Biofilm Formation of *Lacticaseibacillus paracasei subsp. paracasei* Strain ATCC334

**DOI:** 10.3390/ijms21155423

**Published:** 2020-07-30

**Authors:** Jana Al Azzaz, Alissar Al Tarraf, Arnaud Heumann, David Da Silva Barreira, Julie Laurent, Ali Assifaoui, Aurélie Rieu, Jean Guzzo, Pierre Lapaquette

**Affiliations:** 1Université de Bourgogne Franche-Comté (UBFC), AgroSup Dijon, F-21000 Dijon, France; jana_azzaz@hotmail.com (J.A.A.); elissartarraf1@gmail.com (A.A.T.); heumannarnaud@gmail.com (A.H.); david3.dasilva@gmail.com (D.D.S.B.); julie.laurent@u-bourgogne.fr (J.L.); ali.assifaoui@u-bourgogne.fr (A.A.); aurelie.rieu@u-bourgogne.fr (A.R.); 2Muséum National d’Histoire Naturelle (MNHN), Centre National de la Recherche Scientifique, UMR7245, Molécules de Communication et Adaptation des Microorganismes (MCAM), 75005 Paris, France

**Keywords:** Probiotic, *Lacticaseibacillus*, *Lactobacillus*, Resveratrol, polyphenols, biofilm, adhesion

## Abstract

Bacterial strains of the Lactobacillaceae family are widely used as probiotics for their multifaceted potential beneficial properties. However, no official recommendations for their clinical use exist since, in many cases, oral administrations of these bacteria displayed limited beneficial effects in human. Additional research is thus needed to improve the efficiency of existing strains with strong potential. In this context, we assess in vitro the effects of nine polyphenols to stimulate biofilm formation by lactobacilli, a feature enhancing their functionalities. Among these polyphenols, we identify trans-Resveratrol (referred to hereafter as Resveratrol) as a potent inducer of biofilm formation by *Lacticaseibacillus paracasei* (formerly designated as *Lactobacillus paracasei*) ATCC334 strain. This effect is strain-dependent and relies on the enhancement of *L. paracasei* adhesion to abiotic and biotic surfaces, including intestinal epithelial cells. Mechanistically, Resveratrol modify physico-chemical properties of the bacterial surface and thereby enhances *L. paracasei* aggregation, subsequently facilitating adhesion and biofilm development. Together, our in vitro data demonstrate that Resveratrol might be used to modulate the behavior of Lactobacilli with probiotic properties. Combination of probiotics and polyphenols could be considered to enhance the probiotic functionalities in further in vivo studies.

## 1. Introduction

The lactobacilli, and more generally lactic acid bacteria, have been used for centuries for the production of fermented foods [1]. Since bacteria belonging to the *Lactobacillus* genus are extremely diverse, it has been recently proposed a reclassification of this genus into 25 genera taking into account genetic, physiological and ecological criteria [2]. These bacteria are considered as normal beneficial residents of the mammalian gastrointestinal (GI) tract, but also in the oral cavity and vagina, representing in this latter environment the dominant flora [3]. Regarding their relative abundance in the human gut microbiota, bacteria from the Lactobacillaceae family are among the first colonizers and dominant bacteria after birth [4]. In adult, Lactobacillaceae are still present however their proportion is highly variable from one individual to another and they become subdominant, representing about 0.04% of total bacteria in fecal samples [5]. More than 60 species belonging to the Lactobacillaceae family have been retrieved from the human GI tract, including *L. rhamnosus*, *L. casei*, *L. paracasei*, *L. plantarum*, *L. ruminis*, *L. gasseri*, *L. acidophilus*, *L. delbrueckii*, *L. namurensis*, *L. rogosae* and *L. murinus* as predominant species [3,5,6]. Thus, their status of normal GI tract residents and their wide use in traditional and functional foods make Lactobacillaceae strains as bona fide candidates for the development of probiotics. Particularly, the former *L. casei* group, reclassified as the genus *Lacticaseibacillus* [2], consists of three of the most studied Lactobacilli owing to their beneficial properties on the GI tract health. These Lactobacilli are namely *L. casei*, *L. paracasei* and *L. rhamnosus*. Nowadays, the most accepted definition of probiotics is “live microorganisms that, when administered in adequate amounts, confer a health benefit on the host”, but new terms are also emerging to define new concepts in the field of probiotics, such as Live Biotherapeutic Product (LBP) or Next-Generation Probiotic (NGP) [7]. The latter term is often used to refer to non-traditional probiotics such as commensal strains with putative beneficial properties isolated from the gut microbiota (i.e., *Akkermansia muciniphila* or Faecalibacterium prausnitzii). An extensive literature exists describing in vitro and in vivo the multiple beneficial properties on host exhibited by Lactobacillaceae strains, including, but not limited to, the modulation of inflammatory responses, antimicrobial effects to limit pathogens, the modulation of host metabolism or positive effects on the gut/brain axis [8]. Despite there are clinical indications, supported by clinical trials, for the use of some Lactobacillaceae strains in the treatment of human diseases such as antibiotic-associated diarrhea, necrotizing enterocolitis or irritable bowel syndrome (IBS) [9,10,11], no official recommendations for their clinical use exist. In addition, for some human diseases such as the inflammatory bowel diseases (IBDs), comprising Crohn’s disease (CD) and ulcerative colitis (UC), even if some Lactobacillaceae strains are highly effective to cure colitis (mimicking human IBDs) in rodent models, these bacteria displayed very limited effects in human [12,13]. The most promising results to date for probiotics-based approaches have been obtained for induction and maintenance of remission in UC [14,15]. Thus, additional research is required to better understand the probiotic mechanisms of action and to develop next generation probiotics or improve the efficiency of existing strains. 

In this context, our group and others have demonstrated that, compared to the traditional planktonic culture of Lactobacillaceae, the growth of Lactobacillaceae bacteria under biofilm condition can enhance their functionalities by, for instance, protecting them against GI tract stresses (low pH, bile salts), promoting their immunomodulatory properties or stimulating their anti-pathogenic activities [16,17,18]. Biofilms are defined as communities of microorganisms attached to an inert or living surface, enclosed in a self-produced extracellular polymeric substance and thus representing a higher level of organization than single planktonic cells do [19,20]. Since bacteria living under biofilm condition express totally different phenotype traits, compared to the planktonic condition, stimulating the biofilm growth of probiotics might represent an appealing way to improve their efficiency as mentioned above. 

Polyphenols are one of the most diverse group of biologically active plant compounds and are mainly studied for their antioxidants and anti-inflammatory properties that they exert on host cells by acting on a wide range of signaling pathways [21,22,23]. These properties have strong implications for their potential use in the treatment of cancers, neurodegenerative disorders and during aging. Besides their roles in modulating eukaryotic cells, there is now a growing literature on the effects of polyphenols on bacteria activities and even on gut microbiota composition and functions [24]. For decades, many studies pointed out only their antibacterial properties [25], but at lower doses, in the nano- or micromolar range, polyphenols can regulate bacterial metabolisms and processes. The influence of polyphenols on bacterial growth and properties is highly dependent on the polyphenol considered, its dose, but also on the bacterial strain considered. Regarding the Lactobacillaceae family, berries-derived polyphenols have been shown to increase their proportions in the gut microbiota of human healthy subjects [26]. Similar results were obtained using polyphenols derived from cocoa, green tea, grape or red wine extract in various model organisms, suggesting that some polyphenols might act as prebiotics for beneficial bacteria, including those belonging to the Lactobacillaceae family [27,28,29,30]. However, the precise molecular mechanisms by which polyphenols might favor, among others, Lactobacilli in the GI tract remain unclear. 

In the present study, we compared the effects of nine polyphenols on the ability of two *Lacticaseibacillus* strains, *L. paracasei subsp. paracasei* ATCC334 (referred as *L. paracasei* hereafter) and *L. rhamnosus* GG to form biofilm. Among the polyphenols tested, trans-Resveratrol (referred to hereafter as Resveratrol) was the most potent to promote biofilm formation by *L. paracasei* ATCC334, in a strain-dependent manner. Mechanistically, Resveratrol enhanced biofilm formation by modifying the physico-chemical surface properties of *L. paracasei*, hence promoting its adhesion capacities. This work provided new mechanistic insights into how polyphenol, and especially Resveratrol, can change the behavior of *Lacticaseibacillus* bacteria in vitro. Further in vivo studies will be required to decipher whether these changes might impact positively their functionalities as normal resident of the gastrointestinal tract or as probiotics. 

## 2. Results

### 2.1. Low Doses of Polyphenols Modulate Biofilm formation of Lactobacillus Bacteria without Affecting their Growth

We investigated the ability of a panel of polyphenols belonging to various classes, stilbene (Resveratrol), flavonols (Quercetin, Catechecin), hydroxycinnamic acids (p-Coumaric acid, Chlorogenic acid, Caffeic acid) and hydrobenzoic acids (Ellagic acid, Shikimic acid, Protocatechuic acid), to modulate the biofilm formation by two *Lacticaseibacillus* bacteria: *Lacticaseibacillus paracasei* ATCC334 strain and *Lacticaseibacillus rhamnosus* GG strain. These strains have been well characterized for their property to form biofilm in vitro [17,31]. 

Polyphenols are usually added to bacteria or eukaryotic cells at concentrations in the micromolar range, however important discrepancies exist in the literature (from 1 µM to hundreds of µM) [32,33]. Thus, we first evaluated the impact of two doses of each polyphenol, in the micromolar range with a ten-fold difference: 30 µM and 300 µM, on the growth of both *Lacticaseibacillus* strains (Figure A1 and Figure A2). For both strains, there were no significant effects of the nine polyphenols at the low dose (30 µM) on their growth curve, compared to the untreated culture, indicating that these polyphenols were well-tolerated by these bacteria at this low dose. Regarding the high dose of polyphenols (300 µM), Resveratrol and, in a less extent, Quercetin altered the growth of both bacteria (Figure A1A,B and Figure A2A,B) indicating a potential deleterious effect of these polyphenols on the basal metabolism of the *Lacticaseibacillus* bacteria. As a consequence, the next experiences were performed with doses close to 30 µM. Then, we measured the effects of the nine polyphenols on the biofilm formation by *L. paracasei* and *L. rhamnosus*. For this purpose, low dose of polyphenols (30 µM) were added directly in the growth medium of bacteria and biofilm formation on polystyrene support was assessed 24 h later by enumerating biofilm-forming bacteria using a colony forming unit (CFU) assay on agar plates [16]. Three polyphenols: Resveratrol, Catechin and Ellagic acid, significantly stimulated the biofilm formation by *L. paracasei* ATCC334 strain compared to untreated bacteria, with the higher effect observed for the Resveratrol (143,6%) (Figure 1A). By contrast, Quercetin displayed an opposite effect with a reduction in the formation of biofilm by *L. paracasei* ATCC334 (Figure 1A). The other six polyphenols had no significant effects. Regarding the *L. rhamnosus* GG strain, no major positive effects of polyphenol were observed on biofilm formation (Figure 1B), suggesting a strain-dependent effects of Resveratrol, Catechin and Ellagic acid. Only slight, yet significant positive effect of Quercetin was observed. At the opposite, Resveratrol, p-Coumaric acid, Ellagic acid, Caffeic acid, Shikimic acid and Protocatechuic acid significantly tended to reduce the biofilm formation ability of *L. rhamnosus* GG (Figure 1B). Altogether, these results demonstrate potential strain-dependent effects of polyphenols on biofilm formation by *Lacticaseibacillus* and highlight a potential role of Resveratrol to markedly stimulate biofilm formation by *L. paracasei*. Thus, next experiments focused to better characterize this effect of Resveratrol. 

In order to confirm the potential strain-dependent effect of Resveratrol on biofilm formation by *Lacticaseibacillus* strains, we assessed the ability of this polyphenol to modulate biofilm formation by various strains from species belonging to the *L. casei* group (*L. paracasei*, *L. casei*, *L. rhamnosus* and *L. zeae*) (Figure 1C). Beyond the *L. paracasei* ATCC334 strain, Resveratrol treatment also significantly stimulated, to the same extent, the biofilm formation by five other strains of the panel: *L. casei* BL23, *L. casei* VEL12204, *L. rhamnosus* ATCC7469, *L. rhamnosus* ATCC9595 and *L. rhamnosus* VEL12198. At the opposite, Resveratrol treatment can also reduce biofilm formation by some other strains, reinforcing the idea of a strain-dependent effect of this molecule. Of note, Resveratrol treatment (30 µM) did not markedly modified the growth curve of the tested bacteria compared to those obtained with untreated bacteria (Figure A3), except for the *L. paracasei* VEL12194 strain, for which a slight delay was observed upon Resveratrol treatment (Figure A3B in Appendix A). 

### 2.2. Resveratrol Increased the Biofilm Formation by Lacticaseibacillus paracasei by Enhancing Bacterial Adhesion

The human large intestine is covered with a protective mucus layer composed predominantly of mucins proteins secreted by goblet cells [34]. Thus, we used mucin-coated polystyren to mimic the intestinal intraluminal surfaces to visualize and to measure the ability of Resveratrol to enhance the biofilm formation by *L. paracasei* ATCC334 on a biotic surface during 24 h. We first visualized the formation of this biofilm by using the Syto9 probe, labelling all microorganisms in a population, and confocal laser scanning microscopy (Figure 2A). In the three conditions tested: untreated, Resveratrol 10 µM (Resv 10) and Resveratrol 50 µM (Resv 50), L. paracasei ATCC334 formed large flat biofilm structures, but biofilms formed upon Resveratrol treatment looks denser than those generated in the untreated condition (Figure 2A). To objectify these observations, we enumerated the number of living bacteria forming these biofilms at 24 h using the CFU assay. As shown in Figure 2B, a significant higher number of bacteria is retrieved from biofilm formed upon treatment with Resveratrol, with an effect peaking at a dose of 10 µM (167.8%), compared to those obtained from untreated bacteria (set as 100%). The formation of a biofilm can be schematically divided into four phases: (i) adhesion to the abiotic/biotic surfaces, (ii) microcolony formation corresponding to early development of a biofilm architecture, (iii) maturation of the biofilm and (iv) the dispersion (as depicted in Figure 2C) [18]. 

As microbial adhesion is an initial key step on biofilm formation, we analyzed whether Resveratrol treatment can enhance the adhesion of bacteria to a mucin-coated support. Adhesion assay was performed by incubating *L. paracasei* ATCC334 for one hour at 37 °C on a mucin-coated polystyrene support. Adherent bacteria were counted by using the CFU assay (Figure 2D). In line with the increase in biofilm formation by *L. paracasei* observed in Figure 2B upon Resveratrol treatment, a significant increase in the adhesion of *L. paracasei* to the mucin support was observed in bacteria treated with 5 or 10 µM of Resveratrol, compared to untreated bacteria (Figure 2D). This suggests that Resveratrol might stimulate biofilm formation by promoting the adhesion of *L. paracasei* to its support. However, higher doses of Resveratrol (25 and especially 50 µM) tended to reduce the adhesion of *L. paracasei* during this one-hour adhesion assay on mucin. 

### 2.3. Resveratrol Increases Adhesion of Lacticaseibacillus paracasei ATCC334 to Human Intestinal Epithelial Cells without Eliciting an Exacerbated Pro-Inflammatory Response

We next evaluated the effects of the same doses of Resveratrol on the ability of *L. paracasei* ATCC334 to adhere to two human intestinal epithelial cell (IEC) lines: HCT116 (Figure 3A) and HT29 (Figure 3B). For this purpose, Resveratrol was added to the cell culture medium of IECs, one-hour prior *L. paracasei* ATCC334 addition. In both cell lines, Resveratrol treatment significantly enhanced the adhesion of *L. paracasei* ATCC334, with a markedly higher effect for the adhesion to HT29 cells (Figure 3B). Interestingly, if the Resveratrol was added during the biofilm growth of *L. paracasei*, without addition of the polyphenols to host cells, the Resveratrol-treated bacteria remained able to better adhere to IECs compared to untreated bacteria, indicating that Resveratrol might act directly on bacteria to promote their adhesion (Figure 3C). Since an increase adhesion of bacteria to host cells might be detrimental by inducing a pro-inflammatory response [35,36], we checked whether Resveratrol, by promoting adhesion of *L. paracasei* to IECs, might trigger an exacerbated inflammatory response in basal condition (Figure 3D) or in inflammatory condition induced by a lipopolysaccharide (LPS) treatment (Figure 3E). Pro-inflammatory response was assessed by the measure, using ELISA assay, of the secretion of the prototypical pro-inflammatory cytokine IL-8 by HT29 IECs (Figure 3D,E). Interestingly, in basal condition, despite enhanced adhesion property to IECs, Resveratrol-treated *L. paracasei* ATCC334 did not induce more IL-8 upon challenge of IECs compared to untreated control bacteria (Figure 3). Similar observations were made under LPS-induced inflammatory response in HT29 (Figure 3E). Of note, we also checked whether the combination *L. paracasei* ATCC334 and Resveratrol did not induced an excerbated inflammatory response in immune cells by monitoring the secretion of the pro-inflammatory cytokine TNF-α, using the J774 macrophage cell line (Figure A4A,B). 

Thus, these results suggest that, even if Resveratrol enhanced adhesion properties of *L. paracasei* ATCC334 strain to host IECs, it did not elicit a pro-inflammatory response compared to the untreated condition. 

### 2.4. Resveratrol Changes Physico-Chemical Surface Properties of Lacticaseibacillus paracasei ATCC334 Strain

To elucidate the possible mechanism by which Resveratrol might enhance the adhesion properties of *L. paracasei*, and subsequent biofilm formation, we analyzed the surface properties of *L. paracasei* ATCC334 in basal condition and upon Resveratrol treatments. Indeed, the transition from a planktonic lifestyle to an attached state at a surface is a multifactorial process that is particularly determined by chemical and physical properties of the bacterial surface, that displays various electrical charges and hydrophobicity around the bacterial body depending on growth conditions [37]. Knowing the hydrophobic nature of Resveratrol and its ability to interact with numerous biological molecules [38], it might affect surface properties of *L. paracasei* and thereby the interactions of the bacteria with biotic and abiotic supports. To verify this hypothesis, we first performed a global analysis of bacterial surface charges by an electrophoretic mobility assay. As shown in Figure 4A, a significant reduction in the electrophoretic mobility of *L. paracasei* was observed upon treatment with 5 µM Resveratrol (Resv 5) compared to the corresponding untreated bacteria. Even if nonsignificant, a similar trend was observed for higher doses of Resveratrol (Figure 4A). This negative shift in the electrophoretic mobility indicates that Resveratrol treatment increases negative charges at *L. paracasei* surface by revealing new functional groups such as carboxylates or sulfates. In addition, significant changes in conductivity were also measured in Resveratrol-treated bacteria, suggesting that Resveratrol modified their metabolism and the secretion of ions (minerals) which may contribute to the observed increase in the conductivity (Figure 4B). 

Finally, we performed a microbial adhesion to solvents (MATS) assay to characterize the electron–donor/electron–acceptor properties of *L. paracasei* surface upon Resveratrol. In good accordance to the literature [39], untreated *L. paracasei* displayed a relative high affinity to the acidic solvent chloroform, indicating the basic nature of its cell surface (Figure 4C). This affinity tended to be increased by Resveratrol treatments, especially with doses of 10 µM or higher, indicating that Resveratrol favored the basic nature of *L. paracasei* surfaces. Finally, we assessed by MATS the hydrophocity of *L. paracasei* surface by measuring its affinity to the nonpolar solvent hexadecane (Figure 4D). Regarding untreated *L. paracasei*, we confirmed the hydrophilic cell surface properties described in the literature for these bacteria [39], as indicated by the low affinity to hexadecane (less than 5 %) (Figure 4D). Interestingly, Resveratrol treatments induced an increase of about two-fold in the affinity of *L. paracasei* to hexadecane, demonstrating that Resveratrol rendered more hydrophobic the cell surface of the bacteria. This last result is particularly interesting since the hydrophobicity at bacterial surfaces was strongly associated with the ability of bacteria to adhere to abiotic and biotic supports, to form aggregates and to form biofilm [40,41,42,43,44]. 

Altogether, these results demonstrated that Resveratrol-treated *L. paracasei* displayed changes in the physicochemical properties of their surface, especially with a global increase in negative charges, a more basic nature and an increase in their hydrophobicity. These changes might largely contribute to the enhanced adhesion and biofilm formation abilities of Resveratrol-treated *L. paracasei*. 

### 2.5. Resveratrol Promotes Lacticaseibacillus paracasei ATCC334 Aggregation

The hydrophobicity of bacteria cell sufaces is also linked to aggregation, a bacterial lifestyle between the planktonic and biofilm states [45]. In addition, the formation of bacterial aggregates has been proposed to favor biofilm formation, notably by preparing bacteria to switching more rapidly to a biofilm-like phenotype [46]. To evaluate whether Resveratrol can promote *L. paracasei* aggregation, we treated for 1 h 30 min the *L. paracasei* ATCC334 strain with increasing doses of Resveratrol (from 5 to 50 µM) and proceeded directly to microscopic examination of the living cultures. As illustrated by representative micrographs of each condition in Figure 5A, Resveratrol treatments significanlty increased the number of *L. paracasei* bacteria forming aggregates (Figure 5B) and the size of these aggregates (Figure 5C). Aggregation of the bacteria upon Resveratrol treatment was confirmed by performing a sedimentation assay that consisted in measuring the evolution of turbidity during a short time (1 h 30 min) in a static culture [47]. Thus, if a treatment or a stress induced bacterial aggregation, a drop in optical density measured at 600 nm was observed, compared to those observed in a control culture. The significant reduction in turbidity at the top of the culture of Resveratrol-treated *L. paracasei* compared to the value obtained for untreated *L. paracasei* confirmed that aggregation occurred upon Resveratrol treatment (Figure 5D). To conclude, presumably as a consequence of the modification of *L. paracasei* surface properties (Figure 4), Resveratrol induced aggregation of *L. paracasei* and thereby might favor adhesion and biofilm formation (Figure 5E).

## 3. Discussion

Recently, the use of bacteria of the Lactobacillaceae family with the biofilm phenotype has shown to enhance their functionalities. A strategy to enhance the capacity of probiotic strains to form biofilm and consequently their colonization potential could be of first interest. In this study, we identified Resveratrol as an inducer of biofilm formation by *L. paracasei* ATCC334 strain. This effect is strain-dependent and relies on the enhancement of *L. paracasei* adhesion to abiotic and biotic surfaces, which represents the first step in biofilm formation. Resveratrol, by modifying negative charges and promoting a more basic nature and hydrophobicity at bacterial surface, enhanced *L. paracasei* aggregation and subsequently facilitated adhesion and biofilm development.

Resveratrol is more and more often regarded as a beneficial molecule in host–bacterial relationships. On the bacterial side, it is suggested that Resveratrol can have prebiotic-like effects since this polyphenol is able to increase the representation of beneficial bacteria, including those belonging to the Lactobacillaceae family, notably in the context of colitis and obesity [48,49,50,51]. *In vitro*, this modulation can be either positive for some bacterial species (*L. acidophilus*, *L. gasseri*, *L. ruminis*) or negative for some others (*Enterococcus faecalis*, *Escherichia coli* or *Yersinia pseudotuberculosis*), with sometimes strain-dependent effects [52]. In our study, a 30 µM dose of Resveratrol has no effect on the growth rate of *L. paracasei* ATCC334 and *L. rhamnosus* GG, whereas a ten-fold higher dose (300 µM) slows down the growth rate of both species. Resveratrol is also described to display potent antimicrobial activities against some bacterial species, notably by altering energy production, damaging DNA or by altering membrane integrity [53,54]. Thus, we can assume that, in the GI tract, Resveratrol, at a given concentration, might favor some bacterial species while inhibiting the growth of others, and thereby contributing in shaping gut-associated bacterial communities. Finally, Resveratrol can indirectly affect gut bacteria by modulating host processes that, in turn, can regulate bacteria. For instance, Resveratrol can modulate immune responses including processes involved in bacterial clearance, such as xenophagy [55,56].

Current knowledge of the precise mechanisms by which Resveratrol can enhance selectively the representation of Lactobacilli in the GI tract remains largely limited. The data presented in our study emphasize the positive role that Resveratrol can play on *Lacticaseibacillus* by enhancing their aggregation, adhesion, and biofilm formation abilities, presumably by modulating their surface properties. Upon Resveratrol treatment, we observed at *L. paracasei* surface a global increase in negative charges, a more basic nature, and an increase in cell surface hydrophobicity. These results are in agreement with a previous study describing that Resveratrol can modify hydrophobicity on cell surface of Lactobacilli, either by increasing or decreasing it, depending on strains and dose of Resveratrol considered [57]. For instance, a 512 µg/mL dose of Resveratrol (corresponding to about 2243 µM, that is to say 44 times higher than the highest dose used in our study) increases hydrophobicity on cell surface of *L. paracasei* and *L. fermentum*, while decreasing it in *L. plantarum*. This strain-dependent effect of Resveratrol is also illustrated in our study, with an increased biofilm formation by some Resveratrol-treated *Lacticaseibacillus* strains, including the *L. paracasei* ATCC334 strain, whereas a decreased biofilm formation is observed for some others Resveratrol-treated strains of the same group, including three *L. paracasei* strains. Beyond Resveratrol effects on physico-chemical properties of bacterial cell surface, we could not exclude that Resveratrol might act indirectly by, for instance, modifying the production of exopolysaccharides (EPS) substances or by changing the expression profile of cell surface proteins (adhesins, pili) that can be, both of them, involved in adhesion processes [58,59,60,61]. A study using *L. acidophilus* NCFM strain has demonstrated that a 100 µg/mL dose of Resveratrol (corresponding to about 438 µM) stimulates adhesion to intestinal epithelial cells and increases the abundance of some proteins at bacterial surface (pyruvate kinase, 50S ribosomal protein L7/L12, elongation factor P) while decreasing some others (adenylosuccinate synthetase, and 6-phosphofructokinase) [59]. Even if the proteins identified are not belonging to the classical molecular determinants of bacterial adhesion due to their preferential intracellular localization, some of them are considered as moonlighting proteins, playing putative roles at bacterial surface [62]. Changes in the localization of these proteins might be facilitated by the fact that Resveratrol can affect bacterial membrane integrity or can create intracellular stress [53,54,63]. Thus, changes in physico-chemical properties of bacterial surface and modifications of expression levels of surface proteins seems to mediate Resveratrol effects on bacterial adhesion, however further investigations will be required to determine the relative importance of each mechanism. We could assume that these relative contributions of each mechanism might be dose- and strain-dependent and could explain the huge differences observed between various strains in response to Resveratrol treatment. 

Since Resveratrol is still representing a major challenge for food and pharmaceutical industries due to its poor solubility, low bioavailability and possible adverse side effects, doses used to analyze Resveratrol effects in vitro should be consistent with in vivo reachable and tolerable concentration. Pharmacokinetics studies in rodent models demonstrated that in animals receiving *per os* Resveratrol doses from 2 to 240 mg/kg reached a micromolar range concentration in serum [32]. In human, largest tolerable doses used are about 5 g per day and allowed to reach also micromolar range concentration in plasma [38]. Thus, circulating concentrations of Resveratrol observed in rodent models and humans are entirely compatible with Resveratrol doses used in our present study. In addition, colon, by its direct exposition to diet, has been described as a target organ for Resveratrol with higher concentration achievable following oral administration, compared to those obtained in the plasma [64]. These data, demonstrating that Resveratrol can reach at relatively high concentration the colonic environment; suggest, that co-administration of Resveratrol with a probiotic might be effective. Of note, some inter-individual differences can be expected since gut microbiota has been described to convert and to metabolize Resveratrol. As an example, two bacteria in healthy humans, *Slackia equolifaciens* and *Adlercreutzia equolifaciens* have been identified as dihydroresveratrol producers from Resveratrol [65]. It is very likely that depending on the representation of these bacteria, and others involved in Resveratrol conversion, in the gut, Resveratrol availability might greatly differ from one individual to another. This observation can also be true more generally for all other polyphenols since bacteria can transform these compounds in many ways including ring fission, reduction, dihydroxylation, demethylation and decarboxylation [65]. 

To conclude, Resveratrol-treated *L. paracasei* bacteria display enhanced ability to adhere to abiotic and biotic surfaces. Knowing that this ability to adhere to intestinal epithelial cells is one of the criteria used in the selection of probiotic bacteria, formulation of bacteria with Resveratrol might offers an appealing strategy to ameliorate strain characteristics. Moreover, we demonstrated that this increased adhesion contributes to boosting biofilm formation by *L. paracasei*. This positive effect of Resveratrol on biofilm formation represents a novel finding since Resveratrol is essentially described in the literature for its inhibitory activities against biofilm derived from both Gram-positive and Gram-negative bacteria [53]. These inhibitory activities of Resveratrol on biofilm formation are often achieved at higher concentrations than those used in our study, reinforcing the idea that dose selection of Resveratrol used is of primary importance depending on the expected effects and applications and should be carefully tested. Resveratrol dose should also be adapted to the probiotic bacterial strain considered (strain-dependent effects) and to the potential conversion of the molecule by the resident gut microbiota. For this purpose, furthers studies, using model organisms, will be required to ensure the in vivo feasibility of stimulating the functionalities of probiotics by polyphenols. An interesting challenge in the future will be to design and formulate new probiotics, eventually in association with active micronutrients such as Resveratrol, and tailored to integrate individual specific features (resident gut microbiota, clinical context, host genetic). 

## 4. Materials and Methods

### 4.1. Bacterial Strains and Cell Culture

*L. paracasei* ATCC334, *L. paracasei* VEL12194, *L. paracasei* VEL12237, *L. paracasei* LBH1065, *L. casei* BL23, *L. casei* VEL12204, *L. rhamnosus* ATCC7469, *L. rhamnosus* ATCC9595, *L. rhamnosus* VEL12198, *L. rhamnosus* GG and *L. zeae* VEL12211 strains were grown anaerobically without shaking at 37 °C in Man-Rogosa-Sharpe medium (MRS; Condalab) pH 5.8 (adjusted with acetic acid) for biofilm and planktonic cultures (as previously described [17]). *L. paracasei* VEL12194, *L. paracasei* VEL12237, *L. paracasei* LBH1065, *L. casei* VEL12204, *L. rhamnosus* VEL12198, and *L. zeae* VEL12211 strains were kindly provided by L. Bermúdez-Humarán and P. Langella (Micalis Institute, INRA, Jouy-en-Josas, France) and previously characterized in [66]. HCT 116 cells (colonic carcinoma cells), HT-29 cells (colonic carcinoma cells) and J774A.1 macrophages were obtained from ATCC, cultured routinely in Dulbecco’s Modified Eagle Medium (DMEM) GlutaMAX with 10% fetal bovine serum (FBS, Eurobio) and maintained at 37 °C in 5% CO₂ in air. All cell lines have been routinely tested for mycoplasma contamination using the PCR Mycoplasma Test Kit II (PromoKine). Stock solutions of trans-Resveratrol (Sigma), Quercetin (Sigma), Catechin (Sigma), p-Coumaric acid (Sigma), Chlorogenic acid (Sigma), Ellagic acid (Extrasynthese), Caffeic acid (Sigma), Shikimic acid (Sigma) and Protocatechuic acid (Sigma) were prepared in ethanol (50 mM). Bacteria or cells were treated with final concentrations of trans-Resveratrol ranging from 5 to 300 µM. For the other polyphenols, doses of 30 or 300 µM were used.

### 4.2. Growth Curves

Growth curves were performed by measuring Optical Density (OD) using a Tecan infinite 200pro microplate reader (Tecan), with Corning 48 flat bottom transparent polystyrol microplate, with lid and 200 µL per well. Absorbance was measured each hour during 24 h at wavelength 600 nm, at 37 °C, with an orbital shaking (30 s) prior measurement. All strains were seeded at an initial OD of 0.05 (corresponding to 10^7^ CFU/mL) in MRS pH 5.8 in absence (vehicle: ethanol) or presence of the indicated polyphenol at 30 µM or 300 µM, in triplicates. 

### 4.3. Biofilm Formation Assay

24-well polystyrene plates (Costar 3524, Corning Incorporated) were coated with porcine mucin (10 mg/mL, Sigma) in distilled water (200 μL/well; 4 °C, overnight). After discarding the mucin solution, wells were washed twice with a 150 mM NaCl solution and 1 mL per well of fresh MRS, supplemented or not (vehicle: ethanol) with the indicated polyphenol (at concentrations ranging from 5 to 50 µM) and inoculated with 10^7^ colony forming units (CFUs)/mL of a culture in stationary phase of *L. paracasei* ATCC334 or *L. rhamnosus* GG strains. Plates were incubated at 37 °C for 24 h. Cells attached to the well walls were quantified as described previously [16]. After incubation, the medium was removed from each well, and the plates were washed twice in a 150 mM NaCl solution to remove loosely attached cells. We added 1 mL of a 150 mM NaCl solution to each well before repeated pipetting to detach the biofilm, and serial dilutions of biofilm recovered suspension were spotted onto MRS agar plates. Each strain and/or condition was tested in at least three independent experiments, each with three biological replicates.

### 4.4. Adhesion Assay

HCT 116 or HT-29 intestinal epithelial cells were seeded 48 h prior adhesion assay at 4 × 10⁵ cells/ well in 24-well tissue culture plates with DMEM, 10% FBS. *L. paracasei* ATCC334 were cultured overnight in MRS pH 5.8, then washed twice in PBS and resuspended in MRS (adhesion to mucin) or DMEM (adhesion to intestinal epithelial cells) at 10⁷ CFU/mL. Resveratrol (5 to 50 µM) was added directly to MRS (adhesion to mucin, Figure 2D and resveratrol-treated biofilm, Figure 3C), or to DMEM (adhesion to IECs, 1 hour prior *L. paracasei* ATCC334 addition, Figure 2A,B). *L. paracasei* bacteria were added at 10^7^ CFU per well (adhesion to mucin) or at multiplicity of infection (MOI) of 40. After 1 h 30 min, the wells were washed three times with pre-warmed PBS and bacteria were harvested by adding a solution of 1% Triton X-100 in PBS for 10 min. Total bacteria adherent to the mucins/cells were quantified by serial dilution and plating on MRS agar plates. Results were expressed as mean percentage ± standard error of the mean (SEM) of biofilm formation of at least three independent experiments, taken untreated bacteria value as 100%.

### 4.5. Confocal Laser Scanning Microscopy (CLSM)

The structure of *L. paracasei* biofilms was analyzed by confocal microscopy as described previously [16]. *L. paracasei* biofilm, in absence (vehicle: ethanol) or presence of Resveratrol (10 and 50 µM Resveratrol) were formed as described in Section 4.3. Prior to image acquisition, each biofilm was fluorescently labelled with Syto9 probe (labelling all bacteria, Molecular probes), following manufacturer’s instructions. After 10 min of incubation, the sample was placed on the motorized stage of a Leica TCS SP8 (LEICA Microsystems, France) at the DImaCell platform (http://dimacell.fr/), Dijon, France. All biofilms were scanned at 600 Hz at 20× magnification (HC PL APO CS2, 20x/0.75 DRY, NA: 0.75, Refractive Index: 1.00) water immersion objective lens with a 488 nm argon laser set at 0.7% intensity. Emitted fluorescence was recorded within the 500–538 nm range to visualize Syto 9 green fluorescence. Stacks of horizontal plane images (1024 × 1024 pixels) with a z-step of 1 µm were acquired for each biofilm. Three-dimensional projections and sections of a representative biofilm were reconstructed with LAS X software (LEICA Microsystems, France). 

### 4.6. ELISA

HT-29 intestinal epithelial cells or J774-A1 macrophages were seeded 48 h prior adhesion assay at 4 × 10⁵ cells/ well in 24-well tissue culture plates with DMEM, 10% FBS. *L. paracasei* ATCC334 were cultured overnight in MRS pH5.8, under biofilm condition, in presence of absence (vehicle: ethanol) of Resveratrol (5 to 50 µM). Biofilms were then washed twice in PBS and resuspended in DMEM. *L. paracasei* bacteria were added to cells at a MOI of 40 for 4 h with or without a concomittant stimulation with Escherichia coli O127:B8 LPS (100 ng/mL, Sigma). All samples were analyzed in duplicate. After coincubation, cell supernatants were collected and frozen at −80 °C until further analysis. IL-8 (for HT-29 cells) and TNF-α (for J774-A1 cells) concentration in the supernatant were determined by an enzyme-linked immunosorbent assay (ELISA) (Biolegends, San Diego, CA, USA) following manufacturer’s instructions. 

### 4.7. Electrophoretic Mobility and Conductivity

The bacterial suspensions, cultivated in MRS medium and used in its stationary phase, were treated with increasing doses of Resveratrol, from 5 µM (Resv 5) to 50 µM (Resv 50) for 1 h 30 min, and centrifuged for 10 min at 7000× *g*. The pellets were resuspended in 1.5 mM of NaCl with a bacterial concentration of 1 × 10^5^ cells/mL and then washed three times in 1.5 mM of NaCl. The measurements were performed using a ZetaCompact instrument (Cad instrumentation, Les Essarts-le-Roi, France) and electophoretic mobility is expressed in µm/S/V/cm. 

### 4.8. MATS

This partitioning method is based on the comparison between microbial cell affinity to a mono- polar solvent and an apolar solvent. The monopolar solvent can be acidic (electron accepting) or basic (electron donating) but both solvents must have similar Lifshitz-van der Waals surface tension components. Chloroform, an acidic solvent which exhibits negligible basic character when pure, and hexadecane, a strongly basic solvent, were used in this study (Sigma, St Quentin Fallavier, France). *L. paracasei* ATCC334 bacteria were grown overnight in the absence of Resveratrol. Then, the bacterial cultures were exposed for 1 h 30 to different concentrations of Resveratrol (0 µM to 50 µM). According to Pelletier et al. [39], after a centrifugation of 10 min at 7000× *g*, bacteria were resuspended and washed three times in a 150 mM NaCl solution. After a last resuspension of bacteria, OD_400_ was then measured and adjusted to 0.8. This solution (2.4 mL) was mixed by inverting and vortexed for 30 s with 0.4 mL of the indicated solvent. The mixture was allowed to stand for 15 min to ensure complete separation of the two phases before a sample (1 mL) was carefully removed from the aqueous phase and the optical density measured at 400 nm. The percentage of bound cells was subsequently calculated by: % adherence = (1 − A/Ao) × 100 where Ao is the optical density measured at 400 nm of the bacterial suspension before mixing and A is the absorbance after mixing.

### 4.9. Aggregation

The aggregation test was carried out with an overnight planktonic culture of the *L. paracasei* ATCC334 strain. OD_600_ was measured and the appropriate Resveratrol concentration (0, 5, 10, 25, 50 µM) was achieved into each cuvette containing the bacteria at an OD_600_ adjusted to 1. The OD_600_ was then measured after a static culture of 1 h 30 min at 37 °C. After this, 10 μL of the pellet of each cuvette was pipetted and inoculated onto coverslips under an optical microscope for visualization and the number of bacteria per aggregate was counted separately for each condition. 

### 4.10. Statistical Analysis

Data are presented as the mean ± standard error of the mean (SEM). Statistical analyses were performed with GraphPad Prism software (GraphPad Software Inc., San Diego, CA, USA). The non-parametric Mann and Whitney test was used to compare results between conditions. The p-values ≤ 0.05 were considered as statistically significant. 

## Figures and Tables

**Figure 1 ijms-21-05423-f001:**
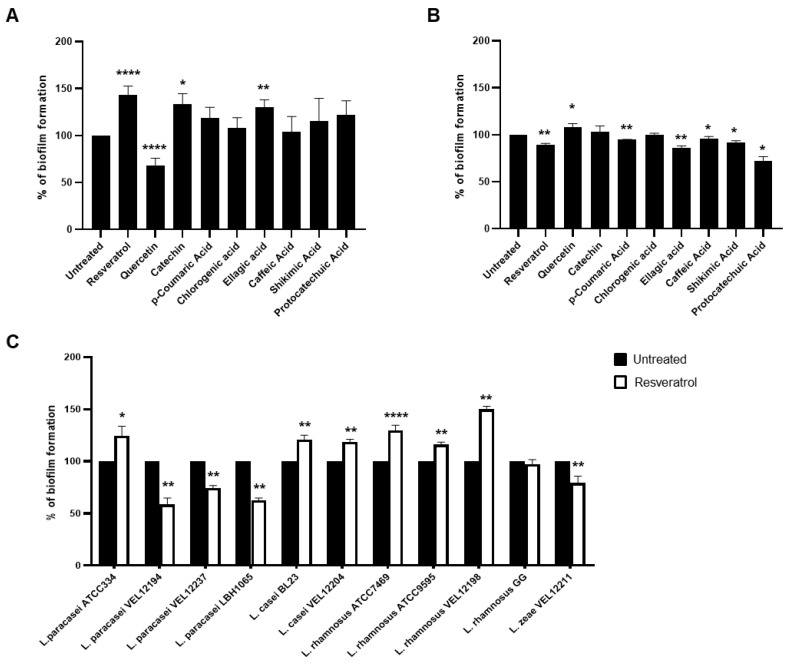
Low doses of polyphenols modulated biofilm formation of *Lacticaseibacillus* bacteria in a strain-dependent manner. (**A**) The ability of *L. paracasei* ATCC334 or (**B**) *L. rhmanosus* GG, untreated or incubated with 30 µM of the indicated polyphenols, to form biofilms was measured after 24 h culture on a polystyrene support by enumeration on Man-Rogosa-Sharpe medium (MRS) agar plates. Data are expressed as mean percentage ± standard error of the mean (SEM) of biofilm formation of at least three independent experiments, taken untreated bacteria value as 100%. * *p* < 0.05, ** *p* < 0.01 and **** *p* < 0.0001 (*versus* untreated). (**C**) The ability of a panel of *Lacticaseibacillus* strains belonging to the *L. casei* group (name indicated below the x-axis), untreated (black bars) or incubated with 30 µM of Resveratrol (white bars), to form biofilms was measured after 24 h culture on a polystyrene support by enumeration on MRS agar plates. Data are expressed as mean percentage ± SEM of biofilm formation of at least six independent experiments, taken untreated bacteria value as 100%. * *p* < 0.05, ** *p* < 0.01 and **** *p* < 0.0001 (*versus* untreated).

**Figure 2 ijms-21-05423-f002:**
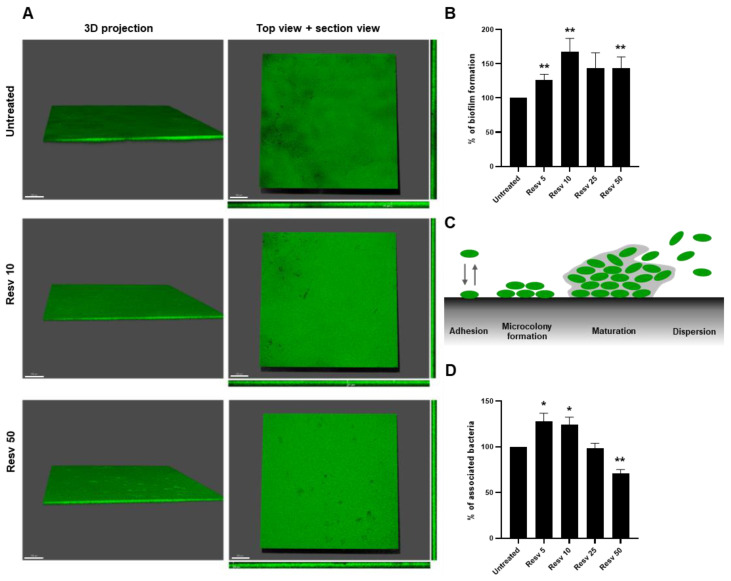
Resveratrol increased the biofilm formation by *L. paracasei* ATCC334 by enhancing its adhesion ability. (**A**) Confocal microscopy images of the biofilm formed by *L. paracasei* grown for 24 h in MRS medium alone (untreated) or in the presence of 10 µM (Resv 10) or 50 µM Resveratrol (Resv 50). The cells in the biofilms were stained with SYTO 9. 3D projections, top and section views are shown. (**B**) Biofilm formation was measured on a mucin-coated polystyrene support by enumeration on MRS agar plates. *L. paracasei* was cultured for 24 h in the absence (untreated) or presence of increasing doses of Resveratrol, from 5 µM (Resv 5) to 50 µM (Resv 50). Data are expressed as mean percentage ± SEM of biofilm formation of at least three independent experiments, taken untreated bacteria value as 100%. ** *p* < 0.01 (*versus* untreated). (**C**) Schematic representation of biofilm formation stages. (**D**) Adhesion of *L. paracasei* to a mucin-coated polystyrene support after a 1 h incubation in the absence (untreated) or presence of increasing doses of Resveratrol, from 5 µM (Resv 5) to 50 µM (Resv 50). Adherent cells were enumerated on MRS agar plates and results are expressed as mean percentage ± SEM of associated bacteria of at least three independent experiments, taken untreated bacteria value as 100%. * *p* < 0.05, ** *p* < 0.01 (*versus* untreated).

**Figure 3 ijms-21-05423-f003:**
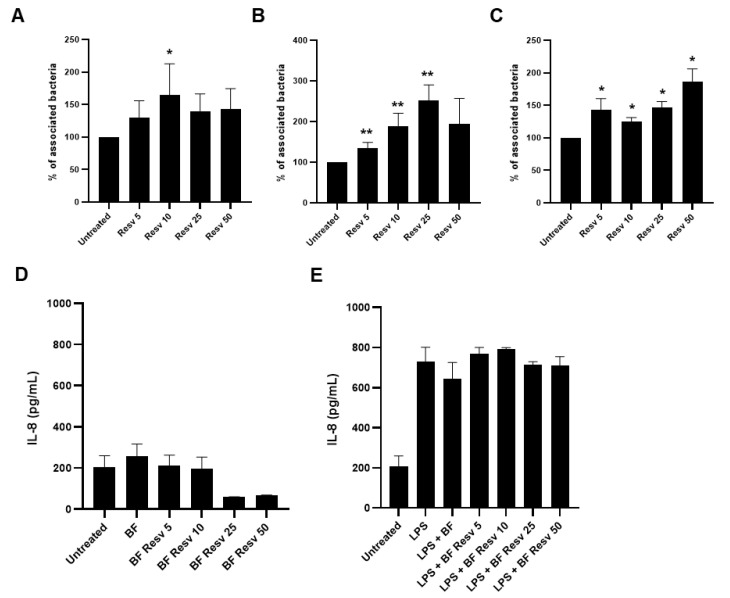
Resveratrol increased adhesion of *L. paracasei* ATCC334 to human intestinal epithelial cells (IECs) without eliciting an exacerbated pro-inflammatory response. (**A**) Adhesion of *L. paracasei* to HCT116 or (**B**) HT-29 IECs untreated or pre-treated for 1 h 30 min with increasing doses of Resveratrol, from 5 µM (Resv 5) to 50 µM (Resv 50). Adherent cells were enumerated on MRS agar plates and results are expressed as mean percentage ± SEM of associated bacteria of at least three independent experiments, taken untreated bacteria value as 100%. * *p* < 0.05, ** *p* < 0.01 (*versus* untreated). (**C**) Adhesion to HT-29 IECs of 24 h biofilm grown *L. paracasei*, in the absence (untreated) or presence of increasing doses of Resveratrol, from 5 µM (Resv 5) to 50 µM (Resv 50). Results are expressed as in A and B panels. (**D**) HT-29 IECs were untreated or treated with biofilm-grown living *L. paracasei* at a MOI of 40 for 4 h. *L. paracasei* were grown under biofilm conditions in the absence (BF) or presence of increasing doses of Resveratrol, from 5 µM (BF Resv 5) to 50 µM (BF Resv 50). IL-8 secretion (pg/mL) was determined by ELISA. Results are expressed as mean ± SEM of at least three independent experiments. (**E**) HT-29 IECs were treated as in D but LPS treatment (100 ng/mL) was added concomitantly to *L. paracasei* treatment.

**Figure 4 ijms-21-05423-f004:**
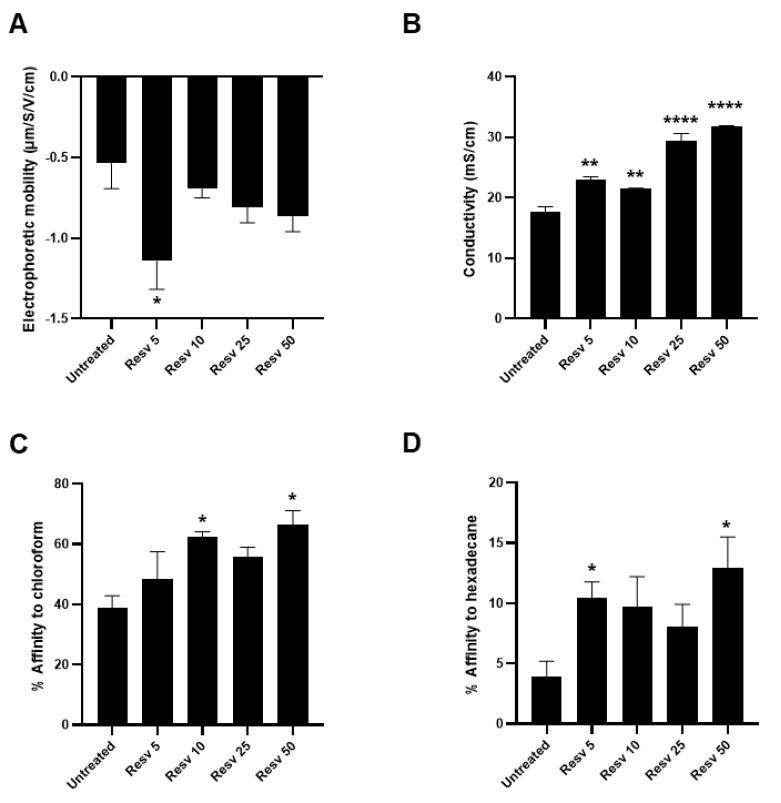
Resveratrol changed physico-chemical surface properties of *L. paracasei* ATCC334 strain. (**A**) Electrophoretic mobility of *L. paracasei* in absence (untreated) or presence of increasing doses of Resveratrol, from 5 µM (Resv 5) to 50 µM (Resv 50). Values (in µm/S/V/cm) represent mean ± SEM of at least five separate measures. (**B**) Measure of conductivity of *L. paracasei* in absence (untreated) or presence of increasing doses of Resveratrol, from 5 µM (Resv 5) to 50 µM (Resv 50). Values (in µm/S/V/cm) represent mean ± SEM of at least five separate measures. (**C**) MATS test, the percentage of adhesion of *L. paracasei,* in the absence (untreated) or presence of increasing doses of Resveratrol, from 5 µM (Resv 5) to 50 µM (Resv 50), to the acidic solvent chloroform or (**D**) to the non-polar solvent hexadecane was measured. Results are expressed as mean percentage of affinity ± SEM of at least three separate experiments. * *p* < 0.05, ** *p* < 0.01 and **** *p* < 0.0001 (*versus* untreated).

**Figure 5 ijms-21-05423-f005:**
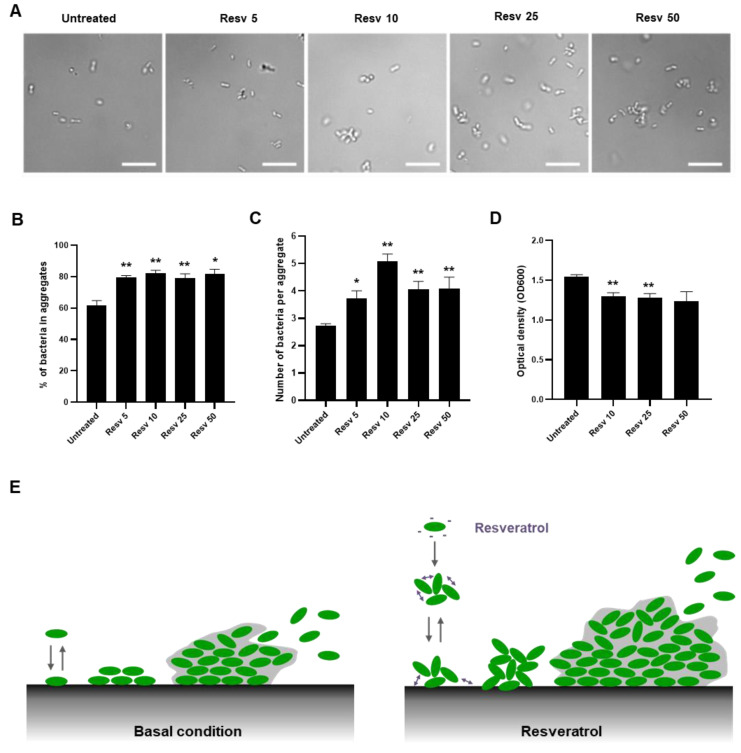
Resveratrol promoted *L. paracasei* ATCC334 aggregation. (**A**) Representative micrographs of live *L. paracasei* treated for 1 h 30 with increasing doses of Resveratrol from 5 µM (Resv 5) to 50 µM (Resv 50) and observed in phase contrast microscopy. In all images, the white scale bar = 10 μm. For each condition, the percentage of bacteria ± SEM forming aggregates is indicated in graph (**B**) and the mean number of bacteria per aggregate ± SEM is shown in graph (**C**). At least, 200 bacteria were counted per condition. * *p* < 0.05, ** *p* < 0.01 (*versus* untreated). (**D**) Sedimentation assay. Optical density at 600 nm (OD_600_) was measured on a static culture of *L. paracasei* untreated or treated with increasing doses of Resveratrol from 5 µM (Resv 5) to 50 µM (Resv 50), after 1 h 30 at 37 °C. Results are expressed as mean optical density at 600 nm ± SEM of at least three separate experiments. ** *p* < 0.01 (*versus* untreated). (**E**) Schematic representation of the effect of Resveratrol treatment on aggregation, adhesion and biofilm formation by *L. paracasei*.

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
