# Peer review of "Resveratrol Favors Adhesion and Biofilm Formation of Lacticaseibacillus paracasei subsp. paracasei Strain ATCC334"

_ijms, 2020, doi:10.3390/ijms21155423_

Round 1
Reviewer 1 Report
The Authors did not provide a sufficient revision of the manuscript.
Author Response
We are sorry but we don't understand why the content of our revised version is not sufficient to satisfy your concerns. Please see the pdf file attached with our point by point response to your previous concerns. For us, we answered to all of it :
- we modified our manuscrit to fit with the new classification of Lactobacillus
- we added and discussed a new experiment to strenghen the fact that resveratrol acts in a strain-dependent manner on bacteria.
- we better explained the polyphenols concentrations used in our study
- we improved phase contrast microscopy images
- we discussed the possibility that EPS might be involved in adhesion process and modified the manuscript accordingly.

Reviewer 2 Report
The manuscript is very interesting and well elaborated. It brings new knowledge about the influence of resveratrol on biofilm formation by probiotic Lactobacillus species. The design of the study is very well, slso the presentation of results and style of writing is satisfactory for the level of the Journal. I have some minor remarks, which however have to be taken into consideration before possible acceptance for publication.
These are following:
Methodology: The authors present the results and discuss the influence of several polyphenols on bacteria and their ability to form a biofilm, but in the methodology section, there is only reference to resveratrol treatment! This should be carefully corrected and all needed data should be included.
Discussion: The first two paragraphs of discussion do not refer to the results and are more appropriate for introduction to give a background of the study. Discussion should aim to explain and discuss the obtained results in a view of a literature, hence please remove or shorten to few sentences this information.
Other: Lines 39-41, the sentence seems not finished…
Author Response
Dear reviewer,
we thank you for your positive feedback to our manuscript.
Please find below a point-by-point response to your concerns. We modified the manuscript accordingly (changes are in red, and sentences removed can be seen with the track changes function) in a new revised version.
These are following:
Methodology: The authors present the results and discuss the influence of several polyphenols on bacteria and their ability to form a biofilm, but in the methodology section, there is only reference to resveratrol treatment! This should be carefully corrected and all needed data should be included.
We apologize for this oversight. We have now added all the necessary details in the materials and methods section (lines 475-479).
Discussion: The first two paragraphs of discussion do not refer to the results and are more appropriate for introduction to give a background of the study. Discussion should aim to explain and discuss the obtained results in a view of a literature, hence please remove or shorten to few sentences this information.
The first two paragraphs of discussion have been edited accordingly to your comment, notably by removing sentences (lines 342-355, 364-366, 370-374).
Other: Lines 39-41, the sentence seems not finished…
Indeed, this is the case, we add the word “criteria” to finish the sentence (line 39-41)
This manuscript is a resubmission of an earlier submission. The following is a list of the peer review reports and author responses from that submission.
Round 1
Reviewer 1 Report
The study of Al Azza et al focused on the effect of resveratrol on biofilm and adhesion properties of L. paracasei ATCC 334. Although the study is very interesting, there are many points to be clarified. According to Int J Syst Evol Microbiol. 2020 Apr 15. doi: 10.1099/ijsem.0.004107, the authors have to change the name of L. paracasei in Lacticaseibacillus paracasei subsp. paracasei.
Major concerns:
1. the authors stated that resveratrol induced biofilm formation in a strain dependent-manner, but the authors considered in their experiments only one strain of L. paracasei (namely ATCC 334) and one strain of L. rhamnosus (namely, L. rhamnosus GG). So, why they state that this is a strain dependent effect? It would be better to test more strain of L. paracasei and L. rhamnosus to get this conclusion.
Line 104-106: could you add the references? Could you bettere explain why you selected these 2 concentrations?
Line 109-113: it is not evident in Fig 1 the deleterious effect of polyphenols.
Line 126-128: still it is not clear why the authors stated “strain dependent”.
Fig 5: the images from phase contrast microscopy are not so clear nor quantitative.
In general, the increased adhesion could depend on EPS production. Is it possible to quantify EPS production after polyphenols exposure? Or authors can comment on this aspect?
Reviewer 2 Report
The manuscript by Jana Al Azzaz et al, entitled “Resveratrol favors adhesion and biofilm formation of Lactobacillus paracasei strain ATCC334” aims to evaluate the impact of polyphenols to improve adhesion and biofilm formation of probiotics, notably of Lactobacillus strains. The authors demonstrated that the effect was different among polyphenols and was also strain-dependent. Resveratrol was notably the best polyphenol able to favor the biofilm formation of a L. paracasei strain and was also able to increase the adhesion of this strain to abiotic and biotic surfaces, notably to epithelial cells. The effects were linked to changes in the surface properties of the bacteria (increase negative charges and hydrophobicity) and to the capacity of resveratrol to promote bacterial aggregation.
The study is well designed and the manuscript well written. However, there are some major concerns the authors should take into account:
Major revision
1) Probiotics can exhibit many beneficial properties and their abilities to adhere to cell surface, to form biofilm and to aggregate are often considered important criteria for their selection, characteristics supposed to favor their better resistance to environmental conditions and survive in the GIT. Although in vitro experiments are key to understand the mechanisms of adhesion and to select probiotic candidates with potential to adhere in vivo, it remains difficult to extrapolate to their beneficial impacts in vivo and there is a lack of correlation in many cases between such in vitro abilities and functional activities in vivo. The authors indeed referred to the paper of Zmora et al (ref 47) about the interindividual variability of probiotics in human trials, which could be relied on individualized gut mucosal colonization. In this paper the authors concluded that the difference of probiotic colonization versus resistance relies on the indigenous gut microbiome composition. The aim of the paper was to evaluate if polyphenols and in particular resveratrol might be used to modulate the behavior of Lactobacillus and enhance their probiotic functionalities. This was not demonstrated in the present work.
To definitively demonstrate the real impact of resveratrol on the adhesion and persistence of the strain in vivo, the authors should compare the kinetics of colonization of the bacteria when administrated in the presence or absence of the polyphenol in animal models (i.e. in healthy mice). This could be achieved by labeling the strains either with antibiotic markers and enumerating the bacteria in the feces or at best in mucosa samples, or by constructing fluorescent or chemiluminescent strains which could be followed by in vivo imaging. The best would be to follow if a beneficial impact could be ameliorated in the presence of resveratrol (i.e. in colitis models). Indeed the authors should This is all the more important since the bioavailability of resveratrol is very low.
2) Why the authors have chosen to work with ATCC334 since this strain do not exhibit a long lasting proof of probiotic property, in order to demonstrate in vivo an impact of adding resveratrol on its functional properties?
3)The impact of resveratrol seems to be strain-dependent since it did not improve the formation of biofilm for rhamnosus GG. It is a pity that the other criteria were not assessed (adhesion to abiotic and biotic surfaces, aggregation…) for the other strains. It would be also interesting to check the strain-specificity by testing another L. paracasei strain. Moreover, as previously reported, LGG seems to adhere through the interaction with pili (Tripahi et al, 2013; bang et al, 2018…), so perhaps a complete different mechanism which is not impacted by resveratrol neither the formation of biofilm. This should be discussed.
4)The authors demonstrated that the increased adhesion of paracasei to epithelial cells was not associated to an induction of pro-inflammatory response. Indeed, treatment of epithelial cells with bacteria alone or in the presence of resveratrol did not increase the level of IL-8 at baseline, neither in the presence of LPS. However, neither the strain alone or in the presence of resveratrol induced an anti-inflammatory effect in LPS-stimulated cells (epithelial cells or macrophages). However, previous published works by the authors, already reported that pro-inflammatory cytokine (TNF-a) production by LPS-activated human monocytoid cells was suppressed by supernatants from different Lactobacillus (other species) cultivated as biofilms but not by planktonic culture supernatants (Aoudia et al, 2016) and this was also reported for the L paracasei ATCC334 used in the present manuscript after THP1 monocytic cell line. How the authors explain these discrepancies?
5) It remains difficult to rely the ability of resveratrol to favor epithelial adhesion without eliciting pro-inflammatory using the J774 macrophage cell line since the authors do not demonstrate the interaction of the bacteria with this type of cells in the presence of the polyphenol. It should have been interesting to unravel if the resveratrol increases or not the phagocytic activity of the macrophages. Description of the results with the two type of cells in the presence or absence of LPS is unclear in the text. Since the work focus on the interaction with IEC, it would be better to focus only on HT29.
6)The aggregation seems not to be very important and highly increased upon resveratrol treatment
Minor comments
- Since the Lactobacillus genus is very heterogenous, leading to aberrant taxonomic situation, a new classification has been proposed in 2018 during the expert LABP meeting (Pot et al, Trends in Food Sci & technol 2019) and published recently ( Zheng et al, Int. J. Syst. Evol. Microbiol 2020). Lactobacillus paracasei and L. rhamnosus, should be renamed as Lacticaseibacillus casei and Lacticaseibacillus rhamnosus,
- The designation of LBP of NGP (line 48) are more restricted to “non traditional probiotics”, i.e. commensal strains isolated from the gut microbiota.
- It should be important in the introduction (line 54-58) to distinguish the effect of probiotics in UC and CD, since a certain number of clinical trials in UC were successful while indeed no clear positive results were obtained in CD (for review see Saez-Lara, et al, BioMed Research International, 2015 and Ghouri et al, Clin Exp Gastroenterol 2014).
- The authors should homogenize the tense of the verb to describe their results (present or past)
- Discussion, line 306, use extra-intestinal instead of systemic
- M&M, bacteria were grown in MRS at 37°C (line 414) but to determine the biofilm formation, plates were incubated at 30°C. Why?
- Line 484: 1 x 105 cells
- Avoid the juxtaposition of numbers at the end of line 496. This is confused
- Line 110: high doses of 300µM